# Economically Important Fruit Flies (Diptera: Tephritidae) in Ghana and Their Regulatory Pest Management

**DOI:** 10.3390/insects16030285

**Published:** 2025-03-10

**Authors:** Elvis Opoku, Muhammad Haseeb, Erick J. Rodriguez, Gary J. Steck, Maria J. S. Cabral

**Affiliations:** 1Plant Protection and Regulatory Services Directorate, Ministry of Food and Agriculture, Accra P.O. Box M37, Ghana; elvis.opoku@mofa.gov.gh; 2Center for Biological Control, College of Agriculture and Food Sciences, Florida Agriculture and Mechanical University, Tallahassee, FL 32307, USA; jessica1.cabral@famu.edu; 3Division of Plant Industry (FDACS/DPI), Florida Department of Agriculture and Consumer Services, Gainesville, FL 32608, USA; erick.rodriguez@fdacs.gov (E.J.R.); g.steck@fdacs.gov (G.J.S.); 4Departamento de Agronomia, Universidade Federal dos Vales do Jequitinhonha e Mucuri, Diamantina 39100-000, MG, Brazil

**Keywords:** agreement, fruit flies, interception, ISPM, management, notification, phytosanitary, regulatory, tephritid, treatment

## Abstract

Fruit flies are major pests and cause severe economic damage to numerous fruits and vegetables in Ghana and beyond. Their impacts include substantial production losses, reduction in incomes, increased poverty, reduced trade volumes of horticultural produce, and a ban or restriction on the export of horticultural produce from developing countries, among others. Despite the progress in their management in Ghana, fruit-fly pests remain problematic. Therefore, a more coordinated nationwide approach and investments in postharvest treatment facilities are required to reduce pest populations, increase trade and incomes, improve the livelihoods of people in the horticultural sector, and stabilize markets for sellers and consumers.

## 1. Introduction

The Economic Community of West African States (ECOWAS), a regional block found in the westernmost part of the African continent, covers about 17% of the continent and consists of Francophone (Benin, Burkina Faso, Cote D’Ivoire, Guinea, Mali, Niger, Senegal, and Togo), Anglophone (Gambia, Ghana, Liberia, Nigeria, and Sierra Leone) and Lusophone (Cape Verde and Guinea Bissau) countries. It was formed in 1975 by the heads of state and governments of member countries to promote economic integration. Agriculture is the primary source of income and livelihood for approximately 35–70% of the population [1,2] and contributes 20–50% to the Gross Domestic Product (GDP) of member countries [3,4,5]. Studies have indicated that the strategic location, coupled with favorable conditions, such as climate and soil when effectively maximized, can help the region to become the next supplier of horticulture products to the world market [3]. The production of high-value horticulture crops (fruits and vegetables) provides excellent opportunities for employment creation, income, and economic growth [6,7], contributes to food and nutrition security [3], and has positive social and environmental effects on exporting countries [8,9]. The horticultural sector has a high potential for reducing poverty due to the intensive use of low-skilled labor in production and post-harvest activities and the high intrinsic value of produce [10]. The West African sub-region contributes to approximately 3.2% of the world production [3]. Ghana plays a significant role in the export of fruits (e.g., mango, pineapple, banana, shea nuts, cashew nut, etc.) and vegetables (e.g., pepper, okra, eggplant, cucumber, luffa, shallot, soybean, squash, gourd, etc.). In 2022, the contribution of agriculture to Ghana’s GDP was approximately 20%, accounting for over 40% of export earnings [11]. Studies have indicated a significant reduction in agricultural exports in 2022 compared to the previous year, resulting in approximately 1% reduction in GDP [3,11].

This reduction in export volumes is partly due to the activities of tephritid fruit flies, which attack all the fruits and vegetables of economic importance in Ghana and the sub-region. Critical among the fruits and vegetables is mango (*Mangifera indica* L.), which is regarded as the “*King of Fruits*”, “*Golden Tree*”, and Ghana’s next “*Gold mine*” [12,13]. It is one of the most important fruit crops grown in tropical and subtropical regions, a highly prized exotic fruit in the European Union market [14,15], and a key component of Ghana’s Non-Traditional Exports (NTEs). West Africa produces 1.4 million tonnes of mango annually and is the 7th largest producer in the world [3]. Despite the enormous strides in mango production made by member countries in the West Africa sub-region, less than 10% of the fresh mangoes are exported (Table 1). It is estimated that 30–40% or more of the mangoes produced annually go to waste due to the incidence of pest infestation (mainly tephritid fruit flies), disease pathogens, and other factors contributing to postharvest losses [16].

This study aimed to assess the effect of regulatory measures on the management of tephritid fruit flies of economic importance in Ghana. It is anticipated that this study will help governments in the ECOWAS sub-region (especially Ghana) appreciate the need to support the regulatory management of tephritid fruit flies as countries seek to increase the production and export of fruits and vegetables.

## 2. Impacts of Major Fruit Flies of Economic Importance

### 2.1. Ghana and the ECOWAS Sub-Region

Sub-Saharan Africa is regarded as the home of many tephritid fruit fly species [19]. The sub-Saharan African tephritid fauna comprises 915 species (from 148 genera) in the wild and cultivated fruits [20,21]. These fruit-infesting tephritid flies are highly polyphagous [20,22,23,24], very destructive, and may either be native (indigenous) to sub-Saharan Africa or invasive [21]. They destroy virtually every horticulture crop (especially fruits and vegetables) of economic importance to the sub-region. The main symptom associated with their damage is the rotting of fruits, which occurs with the help of fruit-rotting bacteria, which the pest introduces into the fruit during oviposition, premature yellowing, and dropping of infested fruits [25,26]. Most fruit-infesting flies on commercially grown fruits and vegetables belong to the genera *Bactrocera*, *Ceratitis*, *Dacus*, and *Zeugodacus* (Table 2) [20,27]. Examples of the most destructive species in Ghana and the sub-region include the Oriental fruit fly (*Bactrocera dorsalis*), Mango fruit fly (*Ceratitis cosyra*), Mediterranean fruit fly (*Ceratitis capitata*), and Melon fruit fly (*Zeugodacus cucurbitae*) [19]. These pests spread rapidly, infest several cultivated (exportable) and wild species of crops, and cause substantial economic losses.

**Table 2 insects-16-00285-t002:** Some tephritid fruit flies of economic importance in Ghana, their hosts, and the extent of their damage.

Species	Origin	Distribution	Host Plants of Economic Importance in the ECOWAS Sub-Region	Quarantine Status	Extent of Damage	References
*Bactrocera dorsalis* (Hendel) 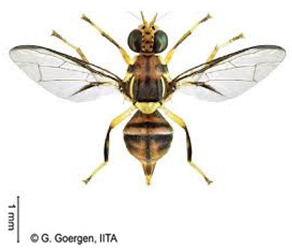 **Some identifying features**: 1. Lateral yellow stripes. 2. Dark hind tibiae. 3. Red-brown scutum with black streaks. 4. Black T-shaped mark on the abdomen.	Invasive (believed to have originated from Sri Lanka but introduced in Africa in 2003)	sub-Sahara Africa, parts of Asia, the Pacific, and Oceania.	Mango, Citrus, Avocado, Banana, Cashew, Cocoa, Coffee, Guava, Papaya, Shea tree, Passion fruit, Pineapple, Sugar apple, Soursop, Watermelon, Tomato, Pepper, Squash, Cucumber, Breadfruit, Jack fruit, Star fruit, African Locust Bean, Yellow mombin, Indian almond, Colocynth, etc.	Quarantine Pests in the European Union, United States, New Zealand, etc.	30–100% crop losses. 80–100% losses in mango and guava. Annual losses in major fruit crops may exceed US$3 billion.	[21,28,29,30,31,32,33,34,35,36,37,38,39,40,41,42,43,44,45,46]
*Ceratitis cosyra* (Walker) 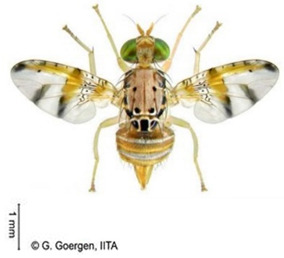 **Some identifying features**: 1. Silver-grey transverse bands on the abdomen. 2. Three separate dark spots on the scutellum. 3. Yellow-black band at the center of the wing. 4. Postpronotal lobe with black spot.	Native	West Africa, parts of Central Africa, East Africa Community (EAC), Southern African Development Community (SADC), and Europe.	Mango (main), Avocado, Cashew, Guava, Soursop, Papaya, Sugar Apple, Shea nut, and Indian almond.	Quarantine Pests in the European Union, United States, and OIRSA.	50–100% crop losses. An average of 20–30% fruit damage in mango.	[19,29,32,45,46,47,48,49,50,51,52]
*Ceratitis capitata* (Wiedemann) 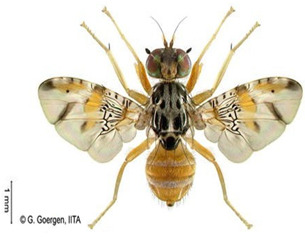 **Some identifying features**: 1. Dark apical spot on the scutellum joined. 2. Silver-grey transverse bands on the abdomen.	Native	West Africa, parts of Central Africa, East African Community (EAC), Southern African Development Community (SADC), Central, and South America, parts of EU, and Australia.	Mango, Citrus, Cashew, Soursop, Sugar apple, Pepper Pawpaw, Coffee, Cucumber, Melons, Cotton, Bitter gourd, Avocado, Passion fruit, Guava, and Tomato.	Not considered a Quarantine pest in the European Union (*EU Regulation 2016/2031*) Quarantine Pests in the United States of America and Canada.	Crop losses may range from 30–100%. 5–15% losses in Coffee. 7–28% losses in Citrus. 27% losses in Guava. 6% losses in Mango. Annual losses are estimated at US$2 billion.	[19,32,47,52,53,54,55,56,57,58,59,60,61,62]
*Ceratitis punctata* (Wiedemann) * 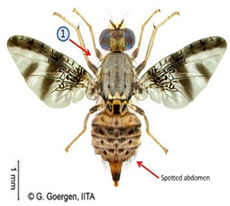 * **Some identifying features**: 1. Three dark spots on the scutellum. 2. Spotted abdomen. 3. Black band at the center of the wing.	Native	West Africa, parts of Central Africa, East African Community, and Southern African Development Community.	Mango, Citrus, Cocoa, Guava, Apple, Soursop, and Sweet berries.	Quarantine Pests in the United States of America and the European Union.	Up to 55% of crop losses are in sweet berries.	[28,32,63,64]
*Ceratitis fasciventris* (Bezzi) 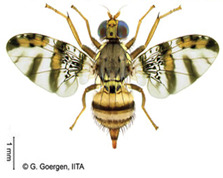 **Some identifying features**: 1. Three separated dark spots on the scutellum 2. Black transverse bands on the abdomen. 3. Black band at the center of the wing. 4. Thick feathering is restricted to the mid-tibiae (males).	Native	West Africa, parts of East Africa Community and Southern African Development Community.	Mango, Avocado, Guava, Soursop, Bitter gourd, Passion fruit, Coffee, Cocoa, Shea nut, and Tomato.	Quarantine Pests in the United States of America and the European Union.	There is an average of 25% crop losses in mango. 80% losses in Guava. Losses of 5% in Citrus.	[36,65]
*Ceratitis quinaria* (Bezzi) 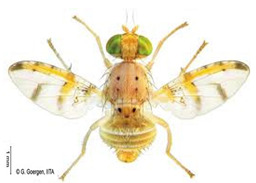 **Some identifying features**: 1. Yellow abdomen with silver-grey transverse bands. 2. Scutellum with five separate black spots. 3. Yellow-black band at the center and anterior margin of the wing.	Native	West Africa (excluding Liberia and Sierra Leone), parts of Central Africa, East Africa Community, and Southern African Development Community.	Mango, Cashew, Guava, Shea nut, and Yellow plum.	Quarantine Pests in Jordan, Israel, the European Union, and the United States of America.	85–95% damage in Yellow plum.	[28,36,46,49,66,67]
*Ceratitis silvestrii* Bezzi 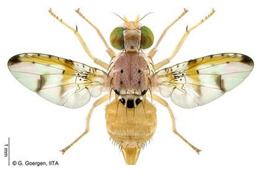 **Some identifying features**: 1. Yellow abdomen with silver-grey transverse bands. 2. Scutellum with three separate black spots. 3. Postpronotal lobe without black spot around the bristle. 4. Prescutellar bristle without dark spot.	Native	Parts of West Africa	Mango, Cashew, Shea nut, White acacia, and Yellow plum.	Quarantine Pests in the United States of America and the European Union.	85–95% crop damage in yellow plum.	[46,67,68,69]
*Ceratitis anonae* Graham 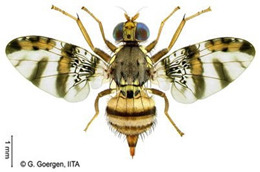 **Some identifying features**: 1. Three separated dark spots on the scutellum. 2. Black transverse bands on the abdomen. 3. Black band at the center of the wing. 4. Both mid tibiae and femora have thick feathering (with no gap at the inner edge of the femur) (males).	Native	Parts of West Africa.	Mango, Soursop, Cocoa, Shea nut, Citrus, Papaya Coffee, Passion fruit, Guava, Avocado, Breadfruit, and Indian almond.	Quarantine Pests in the European Union and United States of America.	Average of 40% crop losses.	[32,36,51,65,70,71,72,73]
*Ceratitis ditissima* (Munro) 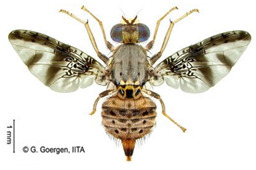 **Some identifying features**: 1. Spotted abdomen. 2. Sickle-shaped dark coloration at the anterior part of the scutum. 3. Black band at the center of the wing. 4. Dark transverse band at tergite-2 of the abdomen.	Native	West Africa, parts of Central Africa, East African Community, and Southern African Development Community.	Mango, Citrus, Cocoa, and Shea nut.	Quarantine Pests in the European Union and United States of America.	1% crop losses in mango.	[32,38,47,70,71,74]
*Dacus bivittatus* (Bigot) 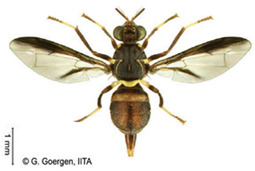 **Some identifying features**: 1. Median and lateral yellow stripes are present on the scutum. 2. Gold transverse band on tergite-2 of the abdomen. 3. Yellow anatergite and katatergite. 4. Costal band complete, large, and extending apically.	Native	West Africa, parts of Central Africa, East Africa Community, and Southern African Development Community.	Mango, Citrus, Papaya, Melon, Watermelon, Cucumber, Squash, Gourd, Tomato, Eggplant, and Luffa.	Quarantine Pests in the European Union and United States of America.	One of the most destructive pests of cucurbits. Data on its yield loss is not readily available.	[32,38,75,76,77,78,79]
*Dacus ciliatus* Loew 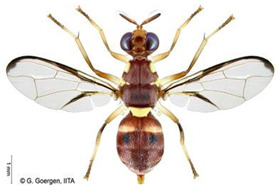 **Some identifying features**: 1. Red-brown scutum with no yellow stripes. 2. Brown anatergite and yellow katatergite. 3. Two dark spots on tergite-3 of the abdomen.	Native	West Africa, parts of (Central Africa, North Africa, East Africa Community, Southern African Development Community, Middle East, and Asia)	Citrus, Cotton, Tomato, Common Bean, Luffa, Snake Gourd, Momordica, Squash, Melon, Cucumber, Watermelon, Blue Passion Fruit, and Bitter Tomato.	Quarantine Pests in the European Union and United States of America.	1.5% of crop damage was recorded in melons under the greenhouse (*with a one-time pesticide application*).	[32,36,39,77,80,81,82]
*Dacus punctatifrons* Karsch 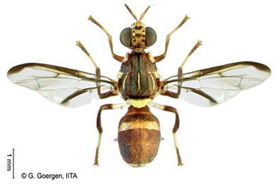 **Some identifying features**: 1. Median and lateral yellow stripes are present on the scutum. 2. Yellow coloration of both anatergite and katatergite. 3. Gold transverse band on tergite-2 of the abdomen. 4. Costal band complete, large, and not extending apically.	Native	West Africa, parts of (Central Africa, East Africa Community, Southern African Development Community, and Middle East)	Citrus, Pepper, Tomato, Watermelon, Cucumber, Squash, Luffa, and Momordica.	Quarantine Pests in the European Union and United States of America.	More than 80% of crop losses are in tomatoes.	[32,46,83,84,85,86,87]
*Dacus frontalis* Becker 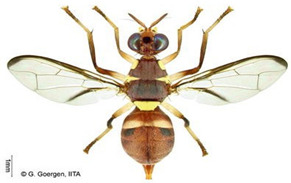 **Some identifying features**: 1. Red-brown scutum with no yellow stripes. 2. Yellow coloration of both anatergite and katatergite.	Native	parts of (West, North, and Central Africa, East Africa Community, Southern African Development Community, and Middle East)	Watermelon, Cucumber, Colocynth, Squash, and Gourd.	Quarantine Pests in the European Union, the United States of America, etc.	Up to 100% yield losses in cucurbits.	[87,88,89]
*Dacus vertebratus* Bezzi 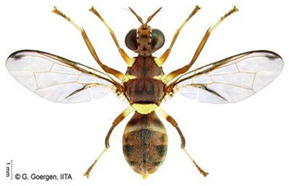 **Some identifying features**: 1. Red-brown scutum with no yellow stripes. 2. Femurs are pale basally and red-brown apically. 3. Slightly expanded spot at the apex of the wing.	Native	parts of (West Africa, Central Africa, East African Community, Southern African Development Community)	Watermelon, Cucumber, Colocynth, Squash, Melon, Tomato and Bitter Gourd.	Quarantine Pests in the European Union, United States of America, New Zealand, etc.	Causes severe yield losses in watermelon	[20,38,74,90,91]
*Zeugodacus cucurbitae* (Coquillett) 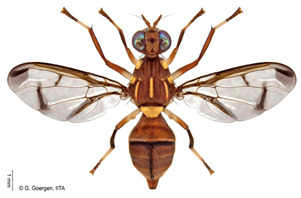 **Some identifying features**: 1. Median and lateral yellow stripes on the scutum. 2. Black T-shaped mark on the abdomen. 3. Wing with a pre-apical cross band.	Invasive (non-dated), native to Central Asia	West Africa, parts of (Central Africa, East African Community, Southern African Development Community), Comoros, French Reunion, Mauritius, Seychelles and the Pacific Islands.	Mango, Cashew, Citrus, Watermelon, Pepper, Melon, Cucumber, Squash, Luffa, Gourd, Bean, Star fruit, Tomato, Okra, and Momordica.	Quarantine Pests in the European Union, United States of America, New Zealand, and Jordan.	95% damage of Bitter gourd 90% damage of Snake gourd	[20,21,32,33,36,38,47,92,93]

**Image source: ©** G. Goergen, Biodiversity Center, IITA.

Tephritid fruit flies remain the worst nightmare for fruit and vegetable farmers in Ghana and the ECOWAS sub-region, attacking virtually all the exported fruits and vegetables. It is estimated that these pests when not adequately managed, could cause yield losses between 30% and 95% depending on the crop, variety, locality, and season [24,28,29]. Such occurrences in the horticultural sector seriously threaten the livelihoods, food and nutrition security, and incomes of millions of people in Ghana [19]. Fruits and vegetables are essential parts of a nutritious and balanced diet for the people in a country affected by persistent high food cost inflation and extreme weather conditions. An example of such fruit is mango, which is eaten across the length and breadth of Ghana. Mango contributes to the food security of the most deprived strata of the population, so populations’ requirement for primary foodstuffs (such as rice, maize, and cassava) is reduced by 40% during mango harvesting season [30]. It is estimated that one mango fruit can provide up to 60% of the daily Vitamin A requirement of people who consume the fruit [31,32]. Unfortunately, mango is one of the fruits severely attacked by tephritid fruit flies in Ghana.

It is estimated that the average mango yield loss associated with fruit flies is 5.65 t/ha with a financial loss of US$3428.97/ha [94]. The yield losses affect the farmers’ incomes and create a deficit for processing industries, which depend on mango as their raw material. These processing industries rely on mango imports from neighboring Burkina Faso and Coté D’Ivoire to operate at total capacity. This situation has severe repercussions on the strength of the local currency. Again, farmers that incur losses due to fruit-fly infestation are unable to pay their creditors and workers, thus affecting the livelihood and incomes of people in the horticultural sector as well as the achievement of the Sustainable Development Goals (SDGs 1, 2, and 3) [33,95].

Tephritid fruit flies are also of quarantine importance. When present in a country, these pests serve as a significant barrier to the export of horticultural produce due to the stringent quarantine restrictions or phytosanitary treatment requirements of trading partners (importing countries). Developing countries (including Ghana) are often unable to meet these requirements and eventually become a barrier to trade. For instance, the United States of America (USA) requires irradiation as the phytosanitary treatment for mangoes. Interestingly, none of the countries in the ECOWAS sub-region have such a facility, which is why Ghana cannot export fresh whole mangoes to the USA. In 2015, Ghana was banned from exporting some selected vegetables to the EU due to pests (including tephritid fruit flies), resulting in the loss of more than US$30 million in export revenue. Similarly, exporters in the sub-region lost approximately €9 million in 2016 due to the interception and confiscation of tephritid fruit fly-infested mangoes at the borders of the European Union (EU) [96]. In Ghana and other developing countries, quarantine restrictions often negatively affect the trading volumes of horticultural produce and create trade deficits, leading to unstable local currency.

### 2.2. Other Countries

Many countries are losing revenue due to fruit fly infestation, the case of Kenya and Mozambique, where both countries lost US$1.9 million and US$17.5 million, respectively, due to *Bactrocera dorsalis* quarantine restriction imposed by South Africa [96,97,98,99,100]. In 2005, Uganda also recorded a 37% decline in fruit exports (estimated at US$436,000) due to tephritid fruit-fly infestation [75]. The economic losses due to tephritid fruit flies are estimated to be more than US$2 billion across Africa [70,101]. The losses incurred due to the US ban on the importation of Spanish clementines following the Medfly (*Ceratitis capitata*) interception were estimated at €300 million [102].

Research has indicated that without management, *B. dorsalis* could cost the citrus industry in China US$25 billion [103]. Similarly, losses associated with *Bactrocera zonata* in the peach industry in China are estimated at US$0.82–3.07 billion [104]. In the Miami-Dade County of Florida, direct losses due to quarantine protocol resulting from an incursion of *B. dorsalis* in 2015 were estimated between US$4–23 million [105]. In Pakistan, job losses associated with tephritid fruit flies were estimated at 726 jobs [106]. Where Sterile Insect Technique (SIT) is used for the management of fruit flies, it is estimated that the annual cost of larval feeding medium for the weekly production of 180 million sterile males of *C. capitata* and *B. dorsalis* is US$1.3 million and US$4.23 million, respectively [107,108].

The other indirect losses from fruit fly infestation include the high cost of surveys, control, and eradication programs [109]. Studies have indicated that rarely more than 1% of mangoes produced in Mali are exported due to fruit fly infestation [110]. Producers, in an attempt to meet the requirements of importing countries, undertake expensive phytosanitary treatments [20]. According to [111], the cost of managing tephritid fruit flies in Ghana ranges between US$688–915.2 per acre per year. In comparison, a return benefit of US$93 for every US$1 investment in the management of fruit flies has been reported [112].

The consumption of infested fruits is also known to cause episodic enteritis in humans, leading to abdominal pains and diarrhea. This became evident after the affected patients consumed the infested fruit of guavas, and *B. dorsalis* larvae were detected in the feces of affected patients [113].

## 3. Pest Management of Tephritid Fruit Flies in Ghana and Trade Barriers

### 3.1. Phytosanitary Measures

The International Standards of Phytosanitary Measures (ISPM #28 with its annexes and ISPM #18) of the International Plant Protection Convention (IPPC) outline some postharvest treatment methods (i.e., vapor heat treatment, cold treatment, and the irradiation of fruits) for the management of tephritid fruit flies and other harmful organisms in international trade and requirements for the use of irradiation as phytosanitary measure, respectively [114,115]. The United States of America, for example, requires irradiation as the treatment for mangoes destined for the US continental market. Unfortunately, none of the member countries in the ECOWAS sub-region (including Ghana) has the facility to undertake these postharvest treatments, thus becoming a barrier to trade. Producers who intend to enter the US market with their fresh whole mangoes may only have to use the “irradiation upon arrival” provision as per the Code of Federal Regulation (7 CFR 305.9) and bilateral agreement between trading parties [116].

Similarly, Articles 4 and 9.9 of the Agreement on Sanitary and Phytosanitary Measures (SPS) of the World Trade Organization (WTO) touch on the equivalence of measures and the recognition of equivalence, respectively [117]. ISPM #24 also provides guidelines for determining and recognizing the equivalence of phytosanitary measures [118]. Ghana and the other ECOWAS member states mainly use the Integrated Pest Management (IPM) approach for the management of tephritid fruit flies (by ISPMs #14, #26) [29,119,120]. Vegetable crop producers who use nets/screen houses for their production attain Pest-Free Area (PFA) status according to International Standards for Phytosanitary Measures (ISPM #4) [121]. The IPM approach provides fruit and vegetable producers with cost-effective methods of reducing yield losses while complying with quality standards. Examples include Cultural Control, Soil Inoculation, Male Annihilation Technique (MAT), Bait Application Technique (BAT), and Use of Biological Agents and Biopesticides [122].

### 3.2. Cultural Control

Cultural control (such as farm sanitation, collection and destruction of fallen and/or infested fruits, harvesting timely depending on the type of fruit, and wrapping of fruits) helps contain and suppress the fruit-fly population [108,122]. Farm sanitation prevents fruit-fly larvae from developing into adult flies [123]. However, some producers (especially non-exporters) often feel reluctant to adopt this strategy due to its labor requirements and management cost [124]. Despite the laborious nature of farm sanitation, it is a very effective fruit fly suppression method and a key component of Integrated Pest Management for tephritid fruit flies [125,126]. In households with few fruit trees, not too tall and within reach, fruits are sometimes wrapped with old newspapers, paper bags, and polythene sheets (especially for bananas) to prevent oviposition [127].

### 3.3. Male Annihilation Technique

With Male Annihilation Technique (MAT), fruit and vegetable crop producers use parapheromones (male lures) to attract and kill male fruit flies [29]. These sexual attractants are volatile components released by host plants and are species-specific. Thus, one requires adequate knowledge regarding the proper identification of the fruit flies in the orchard (Table 2) to determine which lure to use and the species composition in the orchard. For instance, Methyl Eugenol (ME) and Cuelure (CUE) attract several *Bactrocera* species, Zingerone attracts *Bactrocera* species that are not responsive to ME and CUE [21,128,129,130,131,132,133,134,135], and Trimedlure (TML) and Terpinyl Acetate (TA) attract Ceratitis species. Studies have indicated that at least 324 species of male *Bactrocera* are attracted to CUE or raspberry ketone (RK), and 123 species are attracted to ME [136]. The recommended number of traps is usually five (5) per acre. It has been established that more pheromone traps per unit area typically reduce the effectiveness of the control measure, as male flies find it very difficult to locate the source of the odor [137].

### 3.4. Bait Application Technique

The food baits (i.e., Bait Application Technique, BAT) are not species-specific and thus attract both male and female fruit flies [20]. After emergence from their puparium, the adult flies depend on sugar and protein for their growth and development [138,139], and these food sources use bait containing an insecticide that attracts and kills the fruit-fly pests. In Ghana, farmers without imported synthetic baits use palm wine, yeast-sugar mixture [38], and pito mash as food sources for baiting adult fruit flies since they lack.

### 3.5. Biological Control

Using biological agents (e.g., entomopathogenic fungi or predator insects) and botanicals to manage insect pests has also caught up with fruit and vegetable producers in Ghana. Ants such as *Oecophylla longinoda*, where present, reduce fruit oviposition and damage by deterring tephritid fruit flies [140]. Unfortunately, these organisms also tend to secrete formic acids, which make the fruit less attractive, and they cause physical nuisance during fruit harvesting due to their aggressive and highly territorial nature [141,142]. Entomopathogenic fungi (e.g., *Metarhizium anisopliae*) and bio-pesticides (e.g., neem cake) are also applied to the soil to target the larvae/pupae of the pest. Studies have indicated that *M. anisopliae* can persist for over a year when applied to the soil [109,143] and affects the larvae and pupae of *B. dorsalis, Z. cucurbitae*, *C. capitata,* and *C. cosyra* [144]. Recent research in the sub-region has identified other natural enemies (i.e., braconid parasitoids) such as *Fopius caudatus* (Szépligeti), *Psyttalia cosyrae* (Wilkinson), *Psyttalia concolor* (Szépligeti) and *Diachasmimorpha fullawayi* (Silvestri), *Aganapsis* sp., *Tetrastichus giffardianus* Silvestri (Eulophidae, Tetrastichinae)*, Ealata clava* Quinlan (Figitidae, Eucoilinae), *Bracon* sp. (Braconidae, Braconinae)*, Diachasmimorpha* sp. (Braconidae, Opiinae) and *Alysiinae* sp. (Braconidae), etc. [32,52,69,145,146]. However, efficient parasitism on adult flies depends on the natural enemy and the parasite-host ratio. Studies have reported that one *Diachasmimorpha longinoda* female to ten 2–3 instar larvae of *B. dorsalis* increased the parasitism rate to 5.97% [146]. Similar studies have also reported a 57% parasitism rate at a 3:1 ratio of *Fopius arisanus* to *B. dorsalis* [147]. Drew et al. (2005) [35] have also reported that birds and rodents cause a high level of larval mortality by consuming infested fruit.

### 3.6. Chemical Control

The chemical control also forms part of Ghana’s IPM measures used by fruit and vegetable producers. The insecticides (e.g., Spinosad, Malathion, etc.) are often mixed with food substances (such as proteins, sugars, fruit aromas, etc.) and other attractants and then applied as spot sprays or on bands that attract and kill the fruit flies [19]. Both spinosad and malathion are well noted for their low mammalian toxicity and reduced environmental impact on natural enemies, with the latter being affordable and having low levels of fruit fly resistance [148,149,150]. Generally, the combination of IPM approaches has been found to be very effective in managing tephritid fruit flies [151].

### 3.7. Regulatory Pest Management

In Ghana, when the fruits and vegetables are meant for export, the exporter registers with the Plant Protection and Regulatory Services Directorate for the purpose of traceability (in accordance with ISPM 7) [152]. This allows plant health inspectors to monitor farm activities closely to ensure compliance with production protocols and other requirements of the importing country. The produce samples are also subjected to laboratory analysis before harvesting to ensure the fruits are not infested before the producers are allowed to harvest and package for export. The consignments are further subjected to final phytosanitary inspection at the point of exit and only allowed for export when deemed free of fruit fly infestation (ISPM 12) [153].

During the inspection of imported consignments, the exporting country receives notification whenever the importing country intercept pests or the shipment does not comply with the importing country’s phytosanitary requirements [154]. Despite efforts to manage fruit flies, Ghana has received about 180 notifications on tephritid fruit fly interceptions from its trading partners in the EU over the past decade. Interestingly, the situation differs significantly from the sub-region’s other major mango-producing countries (e.g., Burkina Faso, Coté D’Ivoire, Mali, Senegal, etc.) (Figure 1). Generally, tephritid fruit flies have increased interceptions and destruction of fruit exports destined for the EU market [30].

Fruit fly-related seizures and destruction of consignments increased by 23% between 2006 and 2007, and mangoes were the main product in terms of product losses, export volumes, and value [30]. Ghana appears to perform better in managing tephritid fruit flies in the sub-region, especially after 2015, when the country was banned from exporting selected vegetable commodities (Figure 1). The significant reduction in the annual tephritid fruit fly interceptions/notifications by the EU trading partners may partly be attributed to the rigorous regulatory measures of the National Plant Protection Organization (NPPO) and the enhanced diagnostic capabilities of plant health inspectors at the entry or exit points, leading to the regular detection of fruit fly larvae in consignments to be exported, inability of exporters to meet the phytosanitary requirements and the general decline in export volumes. Despite the strict regulatory measures, Ghana occasionally receives tephritid fruit fly interception notifications from its trading partners in the EU. Unfortunately, farms producing fruits and vegetables for the domestic markets are not subjected to these strict management measures, and such farms tend to harbor fruit flies and ruin exporters’ efforts to manage fruit flies. It is estimated that small-scale farmers contribute 80% of the fruits and vegetables to export and local markets [155], and the management of fruit flies is not cooperated with by non-exporters (small-scale farmers). The lack of a nationwide approach to fruit fly management (as in the case of Fall Armyworm) has contributed to some invasive pests becoming established [29].

**Figure 1 insects-16-00285-f001:**
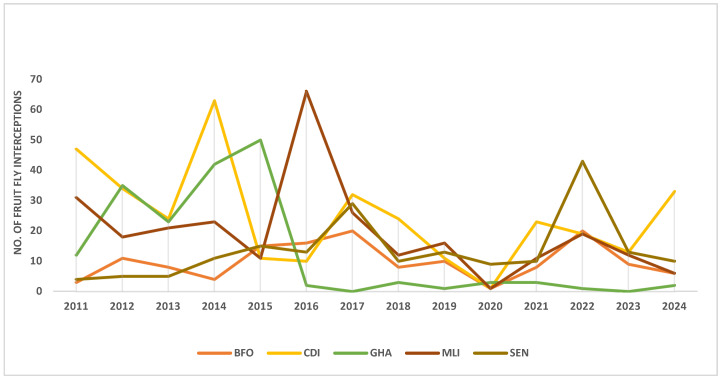
European Union Tephritid fruit flies interception on mangoes from some major exporting countries (BFO = Burkina Faso, CDI = Coté D’Ivoire, GHA = Ghana, MLI = Mali, SEN = Senegal) in the ECOWAS sub-region, 2011– 2024 [156].

## 4. Future Prospects

The ECOWAS sub-region has received funding from international partners for fruit fly research. Most of these studies focused on their management using IPM strategies and achieved some level of success at the local and regional levels. However, fruit flies remain a significant threat to the horticultural sector in Ghana and the sub-region [24,28,29]. This is because many countries have limited resources that they can commit resources for fruit fly management at the national level after project completion. Indeed, there is an urgent need for area-wide management to suppress the population of tephritid fruit flies. Future efforts in controlling the fruit flies need to be on inspection (incoming fruits and vegetables for mature and immature stages of fruit flies), application of sterile insect techniques, and prevention (by controlling movements of infested fruits, harvesting fruits on time, and removal of infested dropped fruit and vegetables from the open fields) [45,46,79,80,156,157,158,159]. In addition, training the grower community and clientele is critical to improving success rates against the fruit flies [160]. Therefore, specific investments and training are necessary for the phytosanitary operations of the National Plant Protection Organization and research sector on using artificial intelligence to detect and screen fruit fly infestations at the points of commodities exit.

## 5. Conclusions

The export of horticulture produce has a positive impact on the Ghanaian economy. However, tephritid fruit flies continue to threaten the food and nutrition security of the people, as well as the incomes of growers and trade. Thus, when left in the hands of the few exporters, their management will have dire consequences for horticultural exports, leading to a budget deficit. All stakeholders in the horticultural sector and the government must adopt a coordinated nationwide approach to fruit fly management, as in the case of Fall Armyworm (*Spodoptera frugiperda*) management in Ghana. An approach where traps will be readily available, accessible, and affordable for all fruit and vegetable fields, and their usage should be continually enforced. The stakeholders, private investors, trading partners, and the government may also engage in long-term investments, such as the construction of treatment facilities within the region to facilitate smooth trade and the economic well-being of producing countries. Indeed, this will also help smallholder farmers keep producing viable crops for consumers who need them and improve people’s food and nutrition security.

## Figures and Tables

**Table 1 insects-16-00285-t001:** The volumes of mangoes produced, consumed, processed, and/or exported (fresh fruit) by some major mango-producing countries in the ECOWAS sub-region, 2022.

Country	Mango Production/yr (Tons)	Fresh Export (Tons)	Processed (Fresh Cuts, Dried, Puree, and Juices) (Tons)	Fresh Consumption (Tons)	Fresh Export %	% Not Utilized
Nigeria	936,934	75	N/A	N/A	0.01	N/A
Mali	350,000	31,000	35,500	25,000	8.86	73.86
Burkina Faso	150,000	8000	38,000	20,000	5.33	56
Cote D’Ivoire	150,000	39,000	1250	15,000	26	63.17
Ghana	99,000	1000	35,000	30,000	1.01	33.33
Senegal	130,000	25,000	3300	20,000	19.23	62.85

N/A (No Available Data). **Source**: [17,18].

## Data Availability

No new data were created or analyzed in this study. Data sharing does not apply to this document.

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
