# Peer review of "Economically Important Fruit Flies (Diptera: Tephritidae) in Ghana and Their Regulatory Pest Management"

_insects, 2025, doi:10.3390/insects16030285_

Round 1
Reviewer 1 Report
Comments and Suggestions for Authors
Summary
This article provides an in-depth analysis of the economic impact of Tephritid fruit flies on Ghana’s horticultural sector, their regulatory management, and the need for improved pest control measures. The study highlights how fruit flies contribute to significant reductions in fruit and vegetable exports, leading to substantial economic losses. It also outlines various pest management strategies, including Integrated Pest Management (IPM) approaches, regulatory interventions, and postharvest treatments. The review is well-written, well-organized, and extensively cited, with distinct sections covering economic importance, key pest species, management strategies, and regulatory frameworks.
Major Concerns
A critical issue in the manuscript is the use of images in Table 2, which were not generated by the authors but copied from previously published papers, books, or online sources. According to the journal's policies, reproducing material without proper attribution and explicit permission from the original copyright holder is not permitted. The authors must either use original images or obtain written consent from the respective sources. If obtaining permission is not feasible, the authors should cite the original sources rather than directly reproducing the images.
Suggestions for Improvement
- Title Revision:
The current title lacks clarity. A more precise and informative title could be:
"Fruit Flies in Ghana: Economic Impacts on Fruits and Vegetables & Strategies for Regulatory Pest Management."
- Comparison with Other Countries:
While the article provides a detailed analysis of pest management challenges in Ghana, similar issues have been faced—and successfully addressed—by other countries. Expanding the discussion to include global best practices and successful interventions would strengthen the paper. For example, incorporating case studies of countries that have effectively solved similar pest problems using integrated pest management (IPM), regulatory policies, biological control, or advanced monitoring techniques could provide valuable insights for policymakers and researchers in Ghana.
- Future Directions & Author Insights:
The conclusion briefly mentions possible solutions, but a more in-depth discussion of future research directions and the authors' insights would enhance the impact of the review. Expanding on potential advancements in pest management strategies, policy recommendations, and emerging technologies would provide a more comprehensive perspective for readers.
Author Response
Summary. This article provides an in-depth analysis of the economic impact of Tephritid fruit flies on Ghana’s horticultural sector, their regulatory management, and the need for improved pest control measures. The study highlights how fruit flies contribute to significant reductions in fruit and vegetable exports, leading to substantial economic losses. It also outlines various pest management strategies, including Integrated Pest Management (IPM) approaches, regulatory interventions, and postharvest treatments. The review is well-written, well-organized, and extensively cited, with distinct sections covering economic importance, key pest species, management strategies, and regulatory frameworks.
Thank you for your time and review of the manuscript. We greatly appreciate this.
Major Concerns. A critical issue in the manuscript is the use of images in Table 2, which were not generated by the authors but copied from previously published papers, books, or online sources. According to the journal's policies, reproducing material without proper attribution and explicit permission from the original copyright holder is not permitted. The authors must either use original images or obtain written consent from the respective sources. If obtaining permission is not feasible, the authors should cite the original sources rather than directly reproducing the images.
Response: As advised, we have requested the permission to reproduce the images from the original author. Hopefully, we will have the permission soon. If not, we will remove the images.
Suggestions for Improvement
- Title Revision:
The current title lacks clarity. A more precise and informative title could be:
"Fruit Flies in Ghana: Economic Impacts on Fruits and Vegetables & Strategies for Regulatory Pest Management."
Response: Thank you. We have revised the title to “Economically Important Fruit Flies (Diptera: Tephritidae) in Ghana and their Regulatory Pest Management”
- Comparison with Other Countries:
While the article provides a detailed analysis of pest management challenges in Ghana, similar issues have been faced—and successfully addressed—by other countries. Expanding the discussion to include global best practices and successful interventions would strengthen the paper. For example, incorporating case studies of countries that have effectively solved similar pest problems using integrated pest management (IPM), regulatory policies, biological control, or advanced monitoring techniques could provide valuable insights for policymakers and researchers in Ghana.
Response: Thank you for the comment. However, the purpose of this review was to focus on Ghana. Therefore, we kept it the same.
- Future Directions & Author Insights:
The conclusion briefly mentions possible solutions, but a more in-depth discussion of future research directions and the authors' insights would enhance the impact of the review. Expanding on potential advancements in pest management strategies, policy recommendations, and emerging technologies would provide a more comprehensive perspective for readers.
Response: As advised, we have added a new section on the Future Prospects as The ECOWAS sub-region has received funding from international partners for fruit fly research over the years. Most of these studies focused on their management using IPM strategies and achieved some level of successes. However, fruit flies still remain a major threat to the horticultural sector in Ghana and the sub-region. This is because governments fail to commit resources for fruit fly management at national level after project completion. There is the need for area-wide management to suppress the population of tephritid fruit flies and investments in the phytosanitary operations of National Plant Protection Organization and research on the use of artificial intelligence in detecting fruit fly infestations at points of exits.
Reviewer 2 Report
Comments and Suggestions for Authors
· The title could be revised for clarity and specificity. For example, "Economically Important Fruit Flies (Diptera: Tephritidae) in Ghana and Their Regulatory Pest Management" would be a more focused and descriptive option.
· In the introduction section, I recommend adding a brief outline of the manuscript’s objectives and the key goals the paper aims to achieve. This will help to better frame the scope of the research.
· In Section 2, it would be helpful to include a list of economically important fruit fly species in Ghana and the broader ECOWAS region. Additionally, incorporating field survey data to substantiate the economic importance of these species would strengthen the argument and provide more context for the reader.
· For Section 3, I suggest merging it with Section 2, as they both address related topics. Regarding Table 2, I recommend relocating the species names to the first column for improved clarity and consistency.
· Sections 4 and 5 could be combined, as the content in these sections is closely related. Merging them would provide a more cohesive and streamlined discussion.
· It would be valuable to add a section on suggestions and future prospects for fruit fly management, with a particular focus on where further research or regulatory improvements are needed. I encourage the authors to place more emphasis on this aspect to enhance the overall impact of the paper.
Author Response
The title could be revised for clarity and specificity. For example, "Economically Important Fruit Flies (Diptera: Tephritidae) in Ghana and Their Regulatory Pest Management" would be a more focused and descriptive option.
Response: Thank you. We have revised the title to “Economically Important Fruit Flies (Diptera: Tephritidae) in Ghana and their Regulatory Pest Management”
In the introduction section, I recommend adding a brief outline of the manuscript’s objectives and the key goals the paper aims to achieve. This will help to better frame the scope of the research.
Response: The objective of this study was to assess the effect of regulatory measures on the management of tephritid fruit flies of economic importance in Ghana. It is anticipated that this study will help governments in the ECOWAS sub-region (especially Ghana) to appreciate the need to support the regulatory management of tephritid fruit flies as countries seek to increase the production and export of fruits and vegetables. This is added in the narrative.
In Section 2, it would be helpful to include a list of economically important fruit fly species in Ghana and the broader ECOWAS region. Additionally, incorporating field survey data to substantiate the economic importance of these species would strengthen the argument and provide more context for the reader.
Response: Examples of the most destructive species in Ghana and the sub-region include Oriental fruit fly (Bactrocera dorsalis), Mango fruit fly (Ceratitis cosyra), Mediterranean fruit fly (Ceratitis capitata) and Melon fruit fly (Zeugodacus cucurbitae) [19]. These pests spread rapidly, infest several cultivated (exportable) and wild species of crops, and cause huge levels of destruction (economic losses).
For Section 3, I suggest merging it with Section 2, as they both address related topics. Regarding Table 2, I recommend relocating the species names to the first column for improved clarity and consistency.
Response: For attention, we kept both sections separately for our readers in Ghana and other parts of the world.
Sections 4 and 5 could be combined, as the content in these sections is closely related. Merging them would provide a more cohesive and streamlined discussion.
Response: For attention, we kept both sections separately for our readers in Ghana and other parts of the world.
Round 2
Reviewer 2 Report
Comments and Suggestions for Authors
1. I still recommend merging Sections 2 and 3, as this aligns with the manuscript's structural requirements. To ensure clarity for readers in Ghana and other regions, sub-sections (e.g., 2.1 and 2.2) could be introduced. The same approach could be applied to Sections 4 and 5.
2. I suggest expanding the "Future Prospects" section by adding more points and relevant references.
Comments on the Quality of English LanguageI recommend that the language in this paper be more succinct and refined.
Author Response
Comments and Suggestions for Authors:
- I still recommend merging Sections 2 and 3, as this aligns with the manuscript's structural requirements. To ensure clarity for readers in Ghana and other regions, sub-sections (e.g., 2.1 and 2.2) could be introduced. The same approach could be applied to Sections 4 and 5.
Response: Thank you for your time and review of this manuscript. We greatly appreciate this. As advised, we have merged the section 2 and 3, and section 4 and 5. Also, sub-sections have been introduced.
- I suggest expanding the "Future Prospects" section by adding more points and relevant references.
Response: We have added several additional points with the relevant references.
Comments on the Quality of English Language
I recommend that the language in this paper be more succinct and refined.
Response: As advised, we have refined the language of the manuscript. Thank you for your time and consideration.